# Psychological and Social Impact of HIV on Women Living with HIV and Their Families in Low- and Middle-Income Asian Countries: A Systematic Search and Critical Review

**DOI:** 10.3390/ijerph19116668

**Published:** 2022-05-30

**Authors:** Nelsensius Klau Fauk, Lillian Mwanri, Karen Hawke, Leila Mohammadi, Paul Russell Ward

**Affiliations:** 1Research Centre for Public Health Policy, Torrens University Australia, 88 Wakefield St, Adelaide, SA 5000, Australia; nelsensius.fauk@torrens.edu.au (N.K.F.); lillian.mwanri@torrens.edu.au (L.M.); 2Institute of Resource Governance and Social Change, Jl. R. W. Monginsidi II, No. 2, Kupang 85221, Indonesia; 3Aboriginal Communities and Families Research Alliance, South Australian Health and Medical Research Institute, Adelaide, SA 5000, Australia; karen.hawke@sahmri.com; 4College of Medicine and Public Health, Flinders University, Adelaide, SA 5001, Australia; leila.mohammadi@health.nsw.gov.au

**Keywords:** WLHIV, psychological and social impact, HIV-affected families, low- and middle-income countries, Asia

## Abstract

Human Immunodeficiency Virus (HIV) infection adds a significant burden to women in Low- and Middle-Income Countries (LMICs), often leading to severe detrimental impact, not only on themselves, but also on their families and communities. Given that more than half of all people living with HIV globally are females (53%), this review seeks to understand the psychological and social impact of HIV infection on Women Living with HIV (WLHIV) and their families in LMICs in Asia, and the interrelationships between one impact and another. A systematic review was conducted to find literature using the following databases: Medline, PsycINFO, CINAL, Emcare, Scopus and ProQuest. Research articles included in this review were selected based on the following inclusion criteria: conducted in LMICs in Asia, published in English language between 1 January 2004 and 31 December 2021, had full text available, involved WLHIV (married and unmarried) and explored the psychological and social impacts of HIV on these women and their families. Critical appraisal tools developed by Joanna Briggs Institute (JBI) were used to assess the methodological quality of the studies, and thematic narrative synthesis was used to analyse the findings. A total of 17 articles met the inclusion criteria. The review showed that HIV has a range of negative psychological consequences on WLHIV, such as stress, fear, worry, anxiety and depression, as well as social impacts on the women and their families, including stigma, discrimination and family separation. The findings indicate the need for targeted interventions—specific to WLHIV—that address the psychological challenges, stigma and discrimination these women and their families face. These interventions should also incorporate education and sustainable support structures for WLHIV and their families.

## 1. Introduction

The Human Immunodeficiency Virus (HIV) and the Acquired Immune Deficiency Syndrome (AIDS) have been a worldwide public health problem for close to four decades [1]. A UNAIDS report reveals an estimated 37.7 million People Living with HIV (PLHIV) worldwide, 1.7 million new diagnoses and 680,000 AIDS-related deaths in 2020 [1,2]. Globally, 53% of the estimated number of PLHIV are women aged 15 and over [2]. In Sub-Saharan Africa (SSA) where the majority (70.5%) of PLHIV reside, Women Living with HIV (WLHIV) represented 63% of HIV infections among adults aged 15 and older [1,2]. The current report also shows an estimated 4200 women aged 15–24 years becoming infected with HIV every week and are twice likely to be living with HIV than men in SSA [2]. In 2020 in Asia and the Pacific region, WLHIV aged 15–24 years and 25–49 years represented 11% and 19%, respectively, of the total number of 5.8 million PLHIV [1]. Globally, around 5000 women aged 15–24 years become infected with HIV every week [1].

It is known that compared to men or other key-affected populations such as transgender populations and men who have sex with men in Low- and Middle-Income Countries (LMICs), women generally face a greater burden of HIV impact [2,3,4,5,6]. LMICs are defined by the World Bank as countries with Gross National Income (GNI) per capita between $1045 or less and $12,695 (7). Countries with GNI per capita of $1045 or less, between $1045 and $4095, and $4096 and $12,695 are, respectively, categorised as low-income countries, lower middle-income countries and upper middle-income countries [7]. Previous studies have reported that WLHIV in LMICs experience considerable psychological challenges, including depression, stress, anxiety and fear due to various concerns facing them in the wake of their HIV diagnosis [1,8,9,10]. WLHIV are also reported to experience a number of social challenges such as stigma and discrimination by others or non-infected people within families, communities, workplaces and healthcare settings, manifesting in various discriminatory and stigmatising attitudes and behaviours [4,8,11,12,13]. A lack of knowledge about how HIV is transmitted and prevented and the fear of contracting HIV through physical, social and healthcare-related contacts have often been reported as the main supporting factors for such stigma and discrimination against WLHIV [12,13,14,15]. For example, these lead to WLHIV being negatively labelled, avoided, rejected and excluded by others within families, communities and healthcare settings due to their HIV-positive status [10,12,13]. Receiving a diagnosis and living with HIV also have negative economic consequences on women and their family through several mechanisms, such as women’s inability to work due to poor physical health condition, loss of employment leading a reduction in or loss of family incomes and increased health expenditure [16,17,18]. Such economic consequences have been reported to also lead to food insecurity within the families of WLHIV and the forced sale of family properties including land and houses to cover healthcare and living expenses [16,18,19,20,21]. HIV diagnosis among women or mothers also causes negative outcomes for the education and well-being of their children [16,22,23], as well as child–mother separation or child-parent conflicts which can have an extremely detrimental effect on the children’s mental health [17,23].

Despite previous studies reporting how HIV affects the lives of WLHIV, to date, there have been no published systematic reviews which specifically address and examine research-based evidence around the psychological and social impacts HIV have on WLHIV and their families in LMICs in Asia and globally [24,25,26,27]. This review provides a better understanding of HIV impact on WLHIV and their families and how HIV impact influences their lives. It is also crucial to gain a greater understanding of the consequences of an HIV diagnosis for women and their families, as well as to identify the drivers behind these consequences given the high burden of disease among WLHIV. Identifying and understanding HIV impact will inform HIV policies and support further research and advocacy around education, targeted evidence-based interventions and healthcare systems that address the specific needs of WLHIV and their families.

## 2. Methods

### 2.1. The Systematic Search of Literature

We performed an initial search of relevant key terms guided by the PICO (Population, Intervention, Comparison and Outcomes), a framework that has been used to inform evidence-based practice [28]. The main search was then drafted and refined in Medline using both Medical Subject Headings (MeSH) terms and keywords, as described in the inclusion criteria section below. We then translated the search into multiple databases, including Medline, PsycINFO, CINAL, Emcare, Scopus and ProQuest.

### 2.2. Inclusion Criteria

Studies were included if they met the following criteria: (i) were published in English and between 1 January 2004 and 31 December 2021, inclusive to capture evidence on the impact of HIV on WLHIV (Appendix A). The choice to cover studies published during this period was due to the reason that antiretroviral therapy became more available for PLHIV in LMICs in 2004) [29]; (ii) were conducted in LMICs in Asia (based on the classification made by the World Bank); (iii) involved WLHIV (married and unmarried); and (iv) explored the psychological and social impacts of HIV on WLHIV and their families (Figure 1). The keywords were used in combination using the Boolean operator system, including AND and OR. An example of a full electronic search strategy in SCOPUS is presented below, and the full key words and search strategy in all databases can be found in Appendix A.

TITLE-ABS-KEY ((*hiv** OR *“Human immunodeficiency virus”* OR *aids*)) AND TITLE-ABS-KEY ((*wives* OR *wife* OR *mothers* OR *female** OR *girl** OR *women* OR *woman*)) AND TITLE-ABS-KEY ((*predictor** OR *“risk factor*”* OR *determinant** OR *“sexual behaviour”* OR *“multiple sex partner*”* OR *extramarital** OR *“sell* sex*”* OR *“transactional sex”* OR *prostitut** OR *“sex work”* OR *condom** OR *”unsafe sex”* OR *“unprotected sex”* OR *knowledge* OR *“social influenc*”* OR *“peer influenc*”* OR *“social norm* “*OR *cultur** OR *sociocultural** OR *socioeconomic** OR *“social environmental* “*OR *socioenvironment** OR *stigma* OR *discriminat** OR *“psychological impact”* OR *“social impact”* OR *stress* OR *distress* OR *depression* OR *“psychosocial impact”*)) AND TITLE-ABS-KEY ((*family** OR *families*)) AND TITLE-ABS-KEY (((*developing* OR *“Less developed”* OR *“low resource*”* OR *disadvantaged* OR *“resource limited”* OR *poor* OR *“low** OR *middle income*”*) W/0 (*countr** OR *region** OR *nation?* OR *area**))) AND (LIMIT-TO (PUBYEAR, *2021*) OR LIMIT-TO (PUBYEAR, *2004*)) AND (LIMIT-TO (LANGUAGE, *“English”*))

### 2.3. Selection of the Studies and Methodological Quality Assessment

847 articles retrieved from the databases and eleven from Google via manual search were collated and imported into Endnote software [30]. After removing 275 duplicates, 583 titles and abstracts were screened by two assessors (Nelsensius Klau Fauk (NKF) and Lillian Mwanri (LM)), further removing 454 articles not meeting the inclusion criteria. The assessment of full texts for the remaining 129 articles led to a further removal of 113 articles not meeting including criteria and one article not meeting methodological quality. The reference listing of the 15 articles was scrutinised, and two additional articles were obtained. The full texts of these two articles were also assessed to determine their eligibility. Seventeen articles fulfilling the inclusion criteria and the methodological quality were finally included in this review (Figure 1). The assessment for methodological quality was performed using critical appraisal tools developed by the Joanna Briggs Institute (JBI) for study design [31]. The methodological quality assessment was performed by two assessors (NKF and LM) and any disagreement between them was resolved through discussion. The appraisal forms for qualitative, cross-sectional and case report studies comprised ten, eight and eight questions, respectively (Appendix A). The questions pertained to the quality of the studies, for which each question received a response of Yes, No, Unclear and Not Applicable.

### 2.4. Data Extraction and Analysis

Guided by Thomas and Harden’s framework [32], a thematic analysis of selected articles was systematically performed as follows: (i) conducting a line-by-line open coding, with one assessor (NKF) extracting free codes from the findings of each article; (ii) developing descriptive themes where the free codes with similarity were organised or grouped together; (iii) reviewing (second assessor LM) the initial descriptive themes, and thoroughly discussing (by both assessors—NKF and LM) any discrepancies; and (iv) finalising the review of descriptive themes and subthemes, with both assessors deciding the final analytical themes for the current study [32]. The final version was agreed upon by all authors following further refinement of the theme and subtheme headings.

## 3. Results

### 3.1. Description of the Included Studies

Table 1 summarises the descriptive characteristics of the 17 included articles. Of the 17 articles, six studies were conducted in India [4,10,33,34,35,36], three in Indonesia [12,37,38], two in Vietnam [39,40], two in Thailand [41,42], two in China [8,43], one in Cambodia [18] and one in Indonesia, India, Thailand and the Philippines [44]. Of the 17 articles, 13 were qualitative studies collecting data through one-on-one and semi-structured or unstructured interviews and focus group discussions [10,12,18,33,34,35,36,37,38,39,40,41,42]. Three articles used quantitative methods (cross-sectional design) [4,8,44] and one was a mixed-method study using a cross-sectional and in-depth interview [43]. A total of 1575 WLHIV participated in these studies, of whom 310 and 1265, respectively, were involved in qualitative and quantitative studies. Several studies involved a combination of WLHIV and Men Living with HIV (MLHIV) [12,42,44] and HIV negative women or women with unknown HIV status [43]; however, only the results regarding WLHIV or the views of WLHIV were included in this review (Table 1). The sample sizes or the number of WLHIV who participated in the qualitative and quantitative studies reviewed varied from 2 to 52 and 90 to 633 people, respectively. The participants were recruited using convenient, purposive or snowball sampling techniques. Five studies involved only married women [4,8,18,38,40], eight involved a combination of married, widowed and single women [10,12,33,36,37,39,41,44] and one study involved single women [42]. Three studies did not report the marital status of the participants [34,35,43]. Participants’ ages ranged from 15 to 60 years old; however, three studies did not report the participants’ ages [35,43,44]. The qualitative studies used content analysis [33,34,35,36,38,43] and thematic analysis [10,12,37,39,40,41,42] approaches for data analysis; however, one study employed a combination of thematic analysis, analysis of episode and the identification of paradigm cases [18]. The quantitative studies used one or a combination of bivariate and multivariate linear or logistic regression models for data analysis [4,8,43,44].

### 3.2. Impact of HIV on WLHIV and Their Families

Two thematic categories, including individual level and family level impact (Figure 2), were identified. Direct quotes from participants were also selected from the studies to support the descriptive synthesis of each theme, as summarised in Table 2. For each of the two categories above, a further synthesis (explanatory phase) was undertaken where different HIV-related impacts were mapped into a simplified Conceptual Model of complex impact (Figure 2) endured by WLHIV and their families. These were subcategorised into two thematic areas (Figure 2), including (i) the psychological impact on WLHIV and (ii) the negative social impact on WLHIV and their families. These themes are discussed in turn below.

**Table 2 ijerph-19-06668-t002:** Qualitative evidence of the impact of HIV on WLHIV and their families.

CATEGORY	SELECTED QUOTES
PSYCHOLOGICAL IMPACT OF HIV ON WLHIV AND THEIR FAMILIES	
**Psychological impact on WLHIV** Distress and fear related to maternal and child health	“I fear my child will be infected with HIV. We are HIV positive and of course there is a high probability for the child of getting infected” Female participant, Vietnam [39].
Depression, stress and anxiety related to the care of the child and the child’d future	“I feel why should I live in this world, or for what am I living? Sometimes I am completely down, depressed emotionally. Sometimes I have feeling of even killing myself or committing suicide.” Female participant, India [34]. “My husband and I fear that we will die early on, when the child is two or three years old only. …. The child who is an orphan will suffer great misery. It will die soon. How will my child live? I cannot imagine what it will be like, I only know that we will be in great misery too.” Female participant, Vietnam [39]. “Every night I cry. I still cry at night until now. When will there be a cure for HIV? How long will I take this medicine? If I cannot be cured, if I die, who will take care of my daughter? She is HIV positive. Who will give her the medicine? My mother is getting older; will my brother’s wife take care of my daughter? This condition encourages me to go to the hospital to get my medicine” Female participant, Indonesia [38]. “I fear what will happen to them when I am not there in this world. I feel my kids are fatherless kids because of the disease, [and] I am a widow.” Female participant, India [34].
Feeling sad, bad and embarrassed and self-blaming	“Yes, madam, every day I am very sad about myself. Because we cannot move freely with everyone, like regular people... At that time [when I was first diagnosed], I felt very bad about myself. I thought, ‘Why should I live this life?’….” Female participant, India [10]. “It is a disease that I would not ever have as I am a good woman and I am only a housewife with one sexual partner, my husband. So, I never thought that I would have HIV/AIDS and I was very shocked the first time I knew my HIV status” Female participant, Indonesia [37]. “I felt people didn’t want to talk to me because I had this disease. I felt embarrassed when I was walking on the street because I used to look so ugly. I was 27 and my weight was 27 kg [60 pounds]. My hair color was black, but I had lost all my hair and looked like a beggar. I felt that people would be afraid of me when they saw me. It was difficult for me to leave the house. I used to leave the house only once a month, to pick up my medication. It was like this for two years...” Female participant, India [10]. “I can’t go to the beauty shop. It makes me feel sad and rejected. … I can talk to my friend when I am sad or hopeless” Female participant, Thailand [45]. “I did not blame anyone for what happened to me. I blamed myself because I had this virus from my previous high-risk behaviors. I blamed myself because I did not listen to my mom’s advice. My mom had prohibited me from having a sexual relationship with my friend” Female participant, Indonesia [37].
Suicidal ideation	“My mental illness affects my physical health; [I] felt like committing suicide.” Another woman shared how mental illness and suicide had touched not only her own life, but that of her husband: “My husband died committing suicide because he too suffered mental illness. I also feel the same way.” Female participant, India [34]. “I feel like taking some sleeping pills or eat some things to end my life.” Female participant, India [34]. “We lived together as a joint family. When they knew about this disease, they [my family] kept away. At that time, I felt very bad, thinking that everyone was healthy. Why has God given me this disease?... They kept away, madam, I felt very bad thinking about how I got this disease. I cried a lot, madam… I felt like I was going to die 149 tomorrow. My husband and I felt horrible and thought about committing suicide. Only because of our children did we not kill ourselves. In our family, nobody is aware of this disease. I have been living with HIV now for 14 years.” *Female participant, India* [10].
**SOCIAL IMPACT OF HIV ON WLHIV AND THEIR FAMILIES**	
**Anticipated and perceived stigma and discrimination among WLHIV**	“Well, I was afraid that they would dislike me because some people heard or saw the news about HIV and understood it while some people didn’t…I didn’t know which of them would accept it and I didn’t want to be bothered by it, so I didn’t tell and that was it.” Female participants, Thailand [42]. “I have not told anyone at work that I’m HIV positive for fear I might lose my job. Because I am afraid that if I tell them that I have HIV, they will remove me from my job. That’s why I did not tell anyone… People would not touch me [if they knew I was HIV-positive]. And they would not even talk in close proximity to an HIV-positive person... They were afraid that if they touch me, they would also get the disease.” Female participants, India [10]. “I thought people would think badly about me, they would say she doesn’t have a husband. She must have done wrong things. That’s the reason she got this. And I thought they would hate me. Because of this fear, I did not tell them. People in the community think that people who are HIV positive made bad choices and that is the reason they are HIV positive.” Female participants, India [10]. “I am afraid that people will *rang kiat* [discriminate] me. Everyone is the same, and they think the same about the illness. It does not matter how many thousand people have HIV/AIDS within the populations of more than 60 millions, I would say that only zero percent will accept people living with HIV/AIDS” Female participant, Thailand [41].
**External stigma and discrimination against WLHIV** From family members ○ Siblings and parents	“Don’t cook for us anymore and don’t use our utensils.” *Female participant, Thailand* [45]. “My father told me to get out of the house. …. For four years, I stayed with my family, in a room. Nobody touched me and nobody talked to me for four years.” Female participant, India [33]. “My mother treated me differently. When I was released from Gandhi Hospital, my parents took me back to their house. One day my mother gave me rice to eat, then my brother’s daughter asked me to feed her. When I was feeding her [with my hands], my mother came and said, “Why are you feeding her your rice?” She said that she would feed her herself. I told my mother that I had not already eaten from the same plate, and that is why I was feeding her. Otherwise, I would not feed her… Sometimes I think that because of my HIV status and my husband’s death, I have lots of problems and I often get fed up with my life. But I have to be alive for my children. I feel sad that everyone is happy, but I am unable to be happy with them... My mother’s sister also has the same feelings towards me. She told my family members to keep my plate, glass, soap—everything—separate. She would say, “Why you are always allowing her to be with you people?” I have suffered a lot this way.” Female participant, India [10]. “After my husband’s death, I lived at my parents’ house. My father, my mother treated me differently. They separated my utensils. They used black tape to mark my plates, spoons, and cups. It was to distinguish that they were mine. I had a small tray, two cups, two plates, two spoons and forks. When they were dirty, I had to clean them by myself using [brand] liquid soap” Female participant, Indonesia [38].
○ From parents-in-law and sisters-in-law	“Since my husband died and I cannot work to provide food for my family-in-law, they treat me with a cold heart. I cannot live there anymore. My mother-in-law sold the house that my husband and I built because it was not yet registered in our names.” *Female participant, India* [40]. “Not only did they blame me, they started beating me! And a few days later, they threw me out of the house. They said ‘You do not have good character so you go away.” Female participant, India [33]. “I was separated from my child (by her sister-in-law). My child slept with her aunty. My eating utensils were given a sign. The relatives of my husband also said to my sisters-in-law: ‘the spoon she used should be separated, you can be infected’. They were nice in front me but felt disgusted about me at the back. They asked my sisters-in-law to chase me and my husband (her husband was HIV-negative) away from the house (the woman and her husband lived together with her sisters-in-law in the same house)” Female participant, Indonesia [12]. “Nine months after my husband’s death, they drove me out from his family house. I’m now renting a room in Denpasar with three of my children while the youngest is still with my parents-in-law” Female participant, Indonesia [37].
From community members (friends, relatives, neighbours)	“Everyone in the neighborhood knows. Stigma is big. My mother–in-law doesn’t care about me, only about my baby. Neighbors visited out of curiosity. Some kept a distance, used bad words, and asked, ‘‘How could an HIV-infected person become a parent?’’ Female participant, India [40]. “I got discrimination in the community where I lived before. If I had touched any foods, then people would not eat those foods. Some (community members) spread information that I am HIV-positive and gossiped about it. I experienced these for about two years” Female participant, Indonesia [12].
From healthcare professionals	“When they knew my HIV status, they shouted at me and did not allow me to sit, even when I was bleeding and weak. They asked other patients to keep away from me. Then they transferred me to a special room. When I gave birth, there was no staff with me.” Female participant, India [40]. “There were nurses who gossiped about my HIV status. They were scared to get close to me or touched me … There was a nurse who told people within the community that I am sick because of this (HIV). She spread information (about his HIV status) within our community that I get HIV” Female participant, Indonesia [12]. “The ANM and staff nurse threw the records on my face and asked me to go to JIPMER for delivery. During that time my membranes ruptured. So I went to JIPMER, throwing my records (in frustration) and delivered there without disclosing my HIV status.” Female participant, India [35].
From employers	“At the beginning of my illness, my face turned black. I did not know about this illness and my husband had already died. I had to work to bring up my two children. But later on I could not do so because my face was black and I was very thin. I was much thinner than I am now. I was asked to leave my job. When I went to apply for any other job, no one took me in and this had an impact on my children.” Female participant, Thailand [41].
Driving factors for stigma and discrimination against WLHIV ○ Fear of contracting HIV and lack of knowledge about HIV	“As most people would not know how we actually get HIV/AIDS, they *rang kiat* [discriminate] women like us. You don’t need to look that far. It is my own sister who has already accepted that I have got HIV. She says she does not *rang kiat* me, but she will be very careful about everything as she still thinks she might get it from me. Like, when she sleeps, she will get another piece of bed cloth to cover where she sleeps. She is very careful. Even water, she will buy her own. What I mean that even if an educated person like my sister is still like this, what about other people? When they know about my HIV status, they will *rang kiat* [discriminate] me for sure.” Female participant, Thailand [41]. “We (the woman and her husband who was also HIV-positive and had died from AIDS) were avoided by nearly all the family members of my husband because they were scared of getting HIV, they did not know how it is transmitted. They thought they would get it if they have physical contact with us. A relative of my husband was the one who spread this misleading information to all the family members of my husband, she told all of them this wrong knowledge, hence they were influenced by what she said. Families and neighbors here are very close to each other, so sensitive information like this (about HIV) can quickly spread and they can easily influence each other and believe it”. Female participant, Indonesia [12].
○ Negative social perception and moral judgement about HIV and PLHIV	“People in community tend to see this disease as *rok mua* [promiscuous disease]. As women, we can have only one partner or one husband. But, for those who have HIV/AIDS, people tend to see them as having too many partners and this is not good. They are seen as *pu ying mai dee* [bad woman who has sex with many men]. And they will be *rang kiat* more than men who have HIV/AIDS.” Female participant, Thailand [41]. “Social perceptions about HIV are very negative, a disease (infection) of people with negative behaviors, such as women who are sex workers, have multiple sex partners or non-marital sex… They perceive HIV as a disgrace for family. Such perceptions influence how other people look at or react towards HIV-positive people … To be honest, I feel uncomfortable with these perceptions”. Female participant, Indonesia [12]. “People have in their heads that HIV transmits because of free sex or sex work, hence many do not respect HIV-positive people. They think we (PLHIV) are immoral because we engage in those immoral behaviors. I can feel it if someone who knows about my (HIV) status and disdains or disrespects me”. Female participant, Indonesia [12].
**Stigma and discrimination against HIV-affected family members** Against HIV-affected family members as a whole	“The day after I learned of my HIV-positive status, my younger sister-in-law escorted all our other family members to go to VCT [voluntary counselling and testing] to get blood tests. She boiled all our bowls and chopsticks. After that, she sold her house and moved away; she doesn’t want to live with my husband and me.” Female participant, India [17]. “People do not join us in eating and they discriminate against my children.” Female participant, Thailand [45].
Against children of WLHIV within communities and schools	“My neighbour would not let her child play with my daughter.” Female participant, Thailand [45]. “Last year my son was at the second grade. He was often sick and had to go to hospital. The teacher told the other children not to play with him because he was sick. He was also isolated from his friends because the teacher placed him at a separate desk. I must stand this discrimination and still let my son go to school. I always warn him not to play with his classmates and not to scratch or bite them.” Female participant, Vietnam [39].
**Family separation** Due to the sickness of the mothers and to get safe life/better home for children with other family	“I had 9 children, but 2 are already dead. I still have 7 children, but only 2 stay with me. When we [she and her husband] were seriously sick, we gave away our children, one to my sister-in-law, one to my sibling. I was at hospital for 8 months. I thought that I was about to die, so all my relatives took my children, one for each”. Female participant, Cambodia [18]. ‘‘Children are at the district and living with my brother in-law there’’ Female participant, Cambodia [18].

#### 3.2.1. Psychological Impact of HIV on WLHIV

Seven studies reported various psychological challenges faced by WLHIV following their HIV diagnosis [8,10,34,37,38,39,43]. Psychological challenges such as depression, stress, worry, anxiety, sadness and embarrassment were the most common experiences among WLHIV [8,10,34,39]. Fear, self-blaming, blaming spouses and feeling shocked, upset and angry due to contracting HIV infection and unacceptability of their HIV-positive status were also experienced by WLHIV [10,37,38,39,43]. These psychological challenges were often triggered by a range of concerns or factors the women felt or faced following an HIV diagnosis. The perceptions that they are good wives, loyal to their spouses and have never had sex with other men, and knowing that they were infected by their spouses were contributing factors for feeling shocked, upset, sad, angry and for their blaming attitude toward their spouses [38]. For some WLHIV, the awareness of their past engagement in unprotected sexual behaviours with multiple partners and the neglect of their parents’ advice toward avoiding sex prior to marriage were mentioned as supporting factors for the self-blaming and sadness they experienced following their HIV diagnosis [37]. An advanced stage of HIV infection, a fear of breach of confidentiality about HIV status, changes in physical appearance and perceiving themselves to be shameful to family were also noted as factors associated with the psychological impact, such as embarrassment, anxiety and depression, for WLHIV [8,10]. A fear of transmitting HIV infection to unborn babies [8,39], concerns about the future of children upon WLHIV untimely death [34,38,39] and a lack of resources to take care of their children [34,39] put additional psychological stress on WLHIV. Similarly, poor economic conditions due to a reduction in work and an increase in family healthcare expenses caused depression and increased fear among WLHIV [38,39]. The lack of social support, social rejection or isolation, perceived stigma, external stigma and discrimination from family members, neighbours and healthcare professionals were also reported in some studies as factors contributing to psychological challenges such as feeling depressed and stressed [10,34]. These factors also led to their inability to cope with the psychological burden they felt, as well as further impacts such as suicide attempts or ideation [10,34].

#### 3.2.2. Social Impact of HIV on WLHIV and Their Families

##### Stigma and Discrimination against WLHIV

Four studies reported women’s feelings of anticipated or perceived stigma following their HIV diagnosis, and they believed that people, including family and community members, would react negatively to their HIV status [10,39,41,42]. For example, they hold the belief that other people will disclose their status and family members will reject them after finding out about the HIV diagnosis [10,41,42]. They also believed that they would be excluded from or denied certain job opportunities because of their HIV-positive status if it is known by others [42]. WLHIV also hold the belief that their children will be stigmatised by other people within communities and denied school admission if their HIV status is known [39]. Such beliefs were based on HIV-related discriminatory and stigmatising attitudes and behaviours previously experienced by other PLHIV and have often led to women isolating themselves to hide the HIV diagnosis from family members [10,39,41,42]. The beliefs were also based on the women’s awareness of various negative perceptions associated with HIV and HIV-infected people [41]. A lower perception of social support from other people within their communities [10,41] was also expressed as a form of anticipated stigma, leading to WLHIV avoiding participation in communal activities.

Eleven studies showed that WLHIV experienced HIV-stigma and discrimination from close family members such as husbands and in-laws, parents and siblings following their HIV diagnosis [4,10,12,33,36,37,38,40,41,43,44]. WLHIV also experienced blaming, verbal insults, avoidance and rejection from husbands and in-laws [4,12,33,40]. WLHIV reported being labelled as sex workers, as well as being beaten and ostracised due to their HIV-positive status; they were also accused by parents-in-law of transmitting HIV to their husbands and were expelled from their marital homes [33,37,38,40,44]. The separation of their eating utensils and refusal of foods and drinks normally shared by in-laws were also experienced by these women [36,38]. Husbands’ occupations, a wider age gap between WLHIV and their husbands and the women’s lower household economic status, financial dependency on their husband and an inability to engage in income generating activities were contributing factors associated with such discriminatory and stigmatising attitudes and behaviours [4]. Women’s inability to engage in income generating activities and the financial dependency on their husbands appeared to also be an explanation for such negative attitudes and behaviours of husbands and in-laws toward them [4,40]. WLHIV also experienced stigma and discrimination from their own families, including from parents and siblings. They were asked to leave home [33,44], excluded from the usual family activities including cooking, sharing foods, drinks and rooms with other family members, and their personal items (e.g., clothes and eating utensils) were separated from those of other family members [4,10,12,33,44]. Having their children taken away from them and a loss of financial support from their family members were also forms of stigma and discrimination faced by these women within their families following the disclosure of their HIV status [12,44].

Seven studies reported HIV stigma and discrimination against WLHIV by friends, neighbours and other community members [4,12,33,40,41,43,44]. Social isolation (e.g., refused entry/excluded from social functions or removed from public establishments) [4,44], eviction from the communities where they lived [33] and refusals by neighbours and relatives to share food and drinks they have touched [12,40] due to the fear of contracting HIV through social contacts were some discrimination incidents that WLHIV faced within communities. WLHIV also experienced physical assaults, negative labelling using discriminatory words such as “HIV carriers” or “she is (HIV) positive” and harassments by other community members [33,40,44]. They were also negatively labelled by neighbours and other community members as sex workers or persons exhibiting bad behaviour [12,40,41].

Similarly, within healthcare settings, WLHIV experienced a range of discriminatory treatments or behaviours committed by healthcare professionals [10,12,35,36,40,41,43,44]. Health professionals exerted the HIV-stigma and discrimination towards WLHIV in forms of criticism, blame, verbal harassment and throwing health records in their face [35,40], as well as by simply avoiding physical contact with them [12,35,36,40]. Refusing to provide healthcare services or leaving them untreated in unattended separated rooms in healthcare facilities [10,12,36,40,44] and unnecessary referrals [10,35] were other forms of HIV-stigma and discrimination coming from healthcare professionals regarding WLHIV. Coercion to undergo HIV testing, to terminate pregnancies, sterilisation and the termination or a loss of private health insurance after HIV diagnosis were other instances of healthcare-related discrimination against WLHIV [44]. WLHIV have also been reported to experience verbal abuse, negative labelling and various accusations from healthcare professionals once their HIV status is known [12,41,43,44]. As a consequence, many WLHIV have chosen to not disclose their HIV status and instead go untreated [35,40], often leading to advanced stages of infection.

Stigma and discrimination towards WLHIV have also occurred within workplace settings, manifesting in the loss of prospects for promotion and unexplained changes in job descriptions due to employers worrying that customers may avoid using the services provided by WLHIV [43,44]. Being asked by employers to leave their jobs due to physical changes following their HIV diagnosis and being rejected by employers due to their HIV-positive status were other HIV-related negative impacts experienced by WLHIV in their workplaces [41].

The fear of contracting HIV, mainly stemming from a lack of knowledge or information about HIV, was reported in some studies as a driver or contributing factor for such stigma and discrimination against WLHIV by others within family, community and healthcare settings [12,37,41]. Other driving factors for stigma and discrimination against WLHIV were negative social perceptions that associated HIV with perceived negative behaviours such as having sex with multiple male partners, engagement in sex work practices and negative moral judgement about WLHIV as the ones who have low moral standing [12,37,41].

##### Stigma and Discrimination against HIV-Affected Family Members and Family Separation

Three studies reported that the husbands and children of WLHIV who were HIV-negative also experienced stigma and discrimination from relatives, other community members, friends and teachers [39,40,44]. For example, husbands’ relatives sterilised (by boiling) utensils (e.g., bowls and chopsticks) that had been used by the women’s husband and children before they could use these themselves due to the fear of contracting HIV, stemming from a lack of HIV knowledge [40]. Children of WLHIV were also negatively affected by their mother’s HIV-positive status. They often faced stigma and discrimination, such as being rejected by their friends and significant others due to their mothers being diagnosed with HIV infection. For example, they were placed at a separate desk by teachers, while other children were told not to play with them at school and within the communities where they lived [39]. Another form of HIV-stigma and discrimination against children of WLHIV was that they were denied school admission or expelled from school due to pressure or complaints from the parents of other students, who themselves feared HIV transmission regarding their children [44].

The forced removal of children by in-laws due to a mother’s HIV diagnosis was another social impact of HIV on HIV-affected families [40,44]. WLHIV also reported voluntarily sending their children to live with their siblings or sisters-in-law/brothers-in-law’s family once they were deemed terminally ill or hospitalised [18]. The main reason for such separations was the fear held by in-laws about mother-to-child transmission and the mothers’ inability to raise their children due to poor physical health and economic conditions [18,40]. Husband–wife separation or abandonment by a husband or partner was another negative impact of HIV on the families of WLHIV, making it hard for women to take care of themselves and their children [18,44].

## 4. Discussion

### 4.1. Summary of Findings

The primary objective of this review was to understand the psychological and social impacts of HIV on WLHIV and their families in LMICs in Asia. A total of 17 studies comprising 13 qualitative studies, 3 quantitative studies and 1 mixed-method study were included in this review. Ten qualitative studies [10,12,33,34,35,36,37,38,41,42] and 3 quantitative studies [4,8,43] focused on exploring the impact of HIV on WLHIV, while 3 qualitative studies [18,39,40] and 1 quantitative study [44] examined the impact on WLHIV and their families.

Psychological challenges faced by WLHIV were presented in one quantitative study [8] and six qualitative studies [10,34,37,38,39,43]. None of these studies explored or reported psychological impacts on the women’s family members. Perceived/anticipated stigma felt by the women following their HIV diagnosis was reported in four qualitative studies [10,39,41,42]. Negative social consequences of an HIV diagnosis in women, such as external stigma and discrimination against them within family, community, workplace and healthcare settings were reported in three quantitative studies [4,43,44] and eight qualitative studies [10,12,33,35,36,37,40,41], while stigma and discrimination towards their family members, especially children within family, community and school settings were reported in 1 quantitative study [44] and 3 qualitative studies [18,39,40]. Other negative social consequences on the women’s families, such as family separation, were reported in 1 quantitative study [44] and 2 qualitative studies [18,40].

### 4.2. Discussion of Psychological and Social Impact of HIV on WLHIV and Their Families

Two levels of HIV impact were identified: the impact on WLHIV at an individual level and on families affected by the women’s HIV diagnosis. Drawing on the reviewed studies and the conceptual framework for the socio-economic impact of the HIV/AIDS epidemic on households [46], an explanatory Conceptual Model for the psychological and social impacts of HIV on WLHIV and their families was developed to visually represent the connection between the emerging themes (see Figure 2).

The model demonstrates that HIV infection results in a range of negative consequences at the individual and familial levels. At the individual level, HIV infection causes psychological challenges including depression, apprehension and anxiety on WLHIV [8,9], often triggered by concerns that the women may face after being diagnosed with HIV. These may include concerns about their poor health, HIV status, stigma, discrimination, family shaming and the future of their children [8,10]. Such concerns reflect psychological responses of WLHIV toward the knowledge about their HIV-positive status and the possibility of various negative consequences they or their families may have to endure [47]. Although articles included in this review do not explore or report psychological impact on families, it is plausible to argue that HIV diagnosis in women can cause psychological challenges for the women’s families, especially children, spouses and parents, as has been reported in some studies in different settings [16,23]. For example, studies in African countries have suggested that HIV-affected children experience a range of mental health issues, often without any sort of support in place [48,49]. HIV infection can also lead to negative social impact, such as stigma and discrimination against WLHIV, reflected in a range of negative attitudes and behaviours towards them, including rejection, avoidance and isolation, which may be imposed by family members, community members, healthcare professionals and employers or colleagues [10,12,41,44]. Stigma and discrimination in LMICs in Asia are also reflected in insults and abuses of WLHIV in many forms by significant others including husbands, partners and in-laws, further exacerbated by sociocultural and patriarchal systems and structures that require wives’ submission to husbands or male partners [33,37,44,50,51]. For example, in some cultures in India and Cambodia, wives are required to submit to their husbands [50,51,52]. Non-submission to husbands would result in sanctions, including beating, divorce, being sent away from home and not receiving financial support [50,51]. Stigma and discrimination toward WLHIV also reflect a lack of support and unacceptance of their HIV status, especially by people around them such as families and friends, which can lead to self-isolation, a lack of access to healthcare and treatment services and the inability to cope with various challenges facing them [8,45,53,54].

A diagnosis of HIV in women can also lead to courtesy stigma toward their family members, especially children, manifesting in rejection or negative treatments by their friends and significant others within communities or school settings [16,23,39,44,45,55]. It can also manifest in denial of school admission and/or discrimination which are serious social consequences of HIV on the children of WLHIV [46,56,57,58]. Thus, the children of WLHIV are an at-risk population and may miss out on crucial education and socialisation with friends at schools and within communities due to stigma and discrimination associated with their mothers’ HIV status [16,17,44]. Another form of HIV-stigma or social impact on the families of WLHIV is forced and voluntary family separation regarding the child and parent, which has also been reported, either due to an inability of WLHIV to take care of their children, a lack of support or unacceptance of women’s HIV status by their own parents, in-laws and spouses or partners [16,18,44].

Stigma and discrimination towards the women themselves and their family are often very damaging and, in many cases, lead to feelings of rejection, hopelessness, shame, anger and fear [8,9,10,17,45]. Notably, the lack of knowledge of HIV transmission and prevention that is reflected in the fear of acquiring HIV infection through interaction with PLHIV is one of the biggest sources of HIV stigma and discrimination [12,41,44,59]. Other sources or divers of HIV-stigma and discrimination toward WLHIV are negative social perceptions about HIV as a promiscuous infection and moral judgement toward PLHIV as people with low moral standing or who behave immorally (e.g., engagement in sex with multiple partners) [12,41]. Stigma and discrimination toward WLHIV and their families seem to also reflect a psychological response of non-infected people toward the knowledge about the existence of PLHIV around them, who themselves may transmit the infection or threaten others’ health status and inevitably their lives [12,13,47].

### 4.3. Implications for Future Practices or Interventions

This systematic review summarises the evidence of a range of HIV negative consequences on an entire family unit when a woman is diagnosed with HIV in a LMIC in Asia. It highlights the psychological and social aspects that need to be addressed in future responses to the HIV issue. It also identifies strategies and interventions that can be used and undertaken to address the psychological and social impacts of HIV on WLHIV and their families.

#### 4.3.1. Addressing Stigma and Discrimination against WLHIV and Their Children

HIV is a stigmatising infection, which varies according to culture and context [60]. Many cultures associate HIV with immorality through an association of the disease with groups deemed “deviant” from social norms, such as homosexual men, commercial sex workers and drug users [61]. In this review, though it was not obvious that WLHIV were linked with any deviant behaviours, their experience of stigma and discrimination was within family, community, healthcare and workplace settings. The children of WLHIV were also stigmatised and discriminated against by their friends and teachers at school, and by other community members just for being related to someone with HIV. These appeared to be influenced by people’s assumption and fear that the children of WLHIV could also be HIV-infected themselves and could transmit the infection to other children [12,13,47]. To respond to HIV-stigma and discrimination effectively, interventions should target not only at the individual or family levels, but also at community and societal levels, including family, community, school, healthcare and workplace settings.

HIV education and awareness of its transmission methods for population groups, families and community members would be a vital component of any intervention, seeking to raise awareness of HIV/AIDS, as well as acceptance of WLHIV [62]. The aim of this would be to reduce stigma and discrimination against WLHIV and to allow their family and community to provide social support [62,63,64]. Key stakeholders in the community would be critical in supporting any HIV intervention, allowing them to lead the community through stages of the intervention [65]. Their opinions and values matter [65]. Similarly, educating and involving both PLHIV who have had their viral load suppressed and been open about their HIV status, and healthcare professionals who provide care to PLHIV, in the delivery of such an intervention would be equally important. These key stakeholders could reinforce the messaging regarding HIV transmission and the safety measures to be adopted. The involvement of these individuals may result in a stronger influence on community members and increase the likelihood of community members receiving the intervention, which can lead to better acceptance of WLHIV and other PLHIV in general and a reduction or elimination of stigma and discrimination towards them within families, communities and other settings [66,67,68,69]. Healthcare system interventions, such as interventions supporting HIV patient self-management and improving patient engagement in HIV care can also be useful strategies to build positive self-perceptions and confidentiality of WLHIV, as well as to address stigma and discrimination against WLHIV [70,71]. These may also lead to reducing internalised stigma which often comes from negative self-perceptions over worry and fear of other people’s reactions.

#### 4.3.2. Addressing Psychological Challenges on WLHIV

Despite improvements in knowledge about HIV and increased access to HIV treatment and support across the world [1], an HIV diagnosis is still found to impose significant psychological challenges on WLHIV, manifested in a range of emotional responses, including stress, depression, sadness, apprehension and anxiety [1,3,10,72]. As such, interventions are needed that tackle psychological challenges faced by WLHIV and their families [73]. Counselling, provided by a trained HIV-counsellor, could be a useful strategy where WLHIV could receive psychosocial support once they are diagnosed with HIV [74]. Counselling in its very nature involves sharing personal experiences and feelings, and therefore counselling services need to consider and account for sex, religious and cultural aspects or norms and values of WLHIV which may increase acceptance and access to the services [75,76]. Family counselling could be a great benefit for WLHIV, as it can increase their understanding and acceptance of the infection, reduce stigma and improve treatment of WLHIV [64,77,78]. Counselling is also reported to increase patients’ adherence to ART, which can improve their physical health [79]. An additional strategy to support WLHIV would be to provide HIV peer support groups, providing them with opportunities to share information, experiences and gain support from others [66,68,69].

### 4.4. Implication for Future Studies

None of the reviewed studies focused on psychological impacts of HIV on the HIV-affected family members of WLHIV, though some studies reported scant information related to the psychological impact on children. Similarly, no studies reviewed looked at stigma and discrimination against family members other than children, nor did they explore the impact of women’s HIV status on the employment of family members. The role that cultural and religious values, norms and thoughts play in exacerbating the impact of HIV on WLHIV and their families in LMICs was also a topic not covered. Also, none of the studies reviewed explored the impact of the women’s HIV-positive status on their access to healthcare services or ART. Future studies that address all of the aspects are recommended in order to develop specific and effective interventions tailored to specific environment and settings, for both WLHIV and their family members. For example, findings of future studies that explore the impact of HIV on the women’s access to ART can be used to improve the HIV-related healthcare system, tailoring HIV care service delivery to address the needs of PLHIV and to support their health and well-being.

### 4.5. Limitation of the Study

This review only included studies conducted in LMICs in Asia. Thus, evidence on the lived experiences of WLHIV in other LMICs outside Asia and in developed countries about the impact of HIV on themselves and their families was not covered in this review. To build a comprehensive picture of HIV impact facing WLHIV and their families, future literature reviews that include studies on this topic conducted in other LMICs and in developed countries are also recommended. Another limitation is that this review included English peer review literature only, and thus we may have missed studies on this topic which are reported in other languages.

## 5. Conclusions

The Conceptual Model shown in this review presents evidence that an HIV diagnosis brings significant, persistent and negative psychological and social consequences for WLHIV and their families. This model may assist in designing interventions that address psychological and social needs to ensure that WLHIV and their families can achieve the best quality of life, as well as the support they need to thrive even in the face of an HIV diagnosis.

## Figures and Tables

**Figure 1 ijerph-19-06668-f001:**
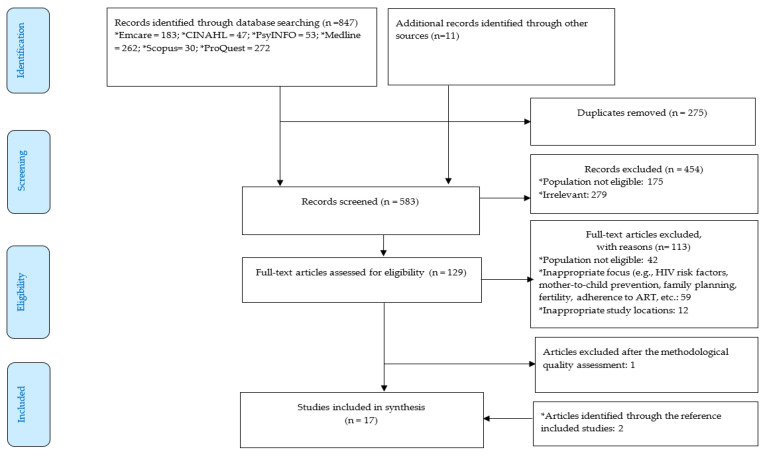
PRISMA Flow diagram of systematic literature search: records identified, screened, eligible and included in the review.

**Figure 2 ijerph-19-06668-f002:**
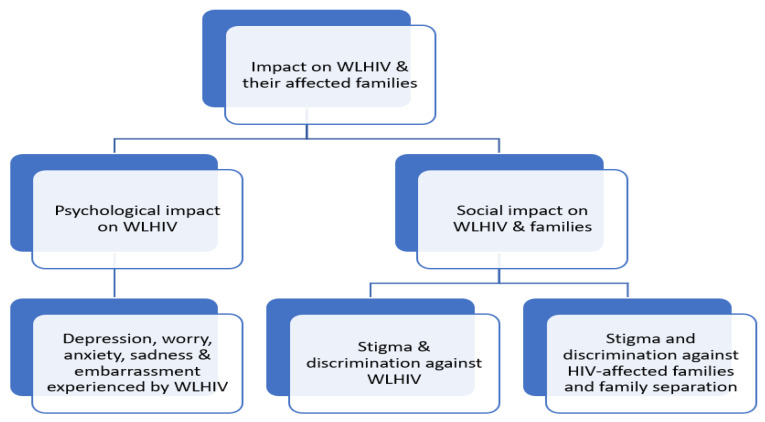
A simplified conceptual model for the psychological and social impacts of HIV on individuals and families.

**Table 1 ijerph-19-06668-t001:** A description of the included studies in alphabetical order.

Author/Year	Study Location	Study Design/Study Aim	Number of Participants/Type of Participants	Analysis	Main Themes of the Impact HIV on WLHIV and Their Families
1. Azhar, 2018 [10]	India	(i) Qualitative design (ii) Methods: In-depth interview (iii) Aim To explore how stigma, social isolation, and depression affect cisgender women living with HIV in Hyderabad, India	(i) 16 WLHIV (ii) Marital status: Majority: widowed Some: married (iii) Participants’ age: Between 18 to 50 years old Mean age: 37.25 years (iv) Participants were recruited using purposive and snowball sampling techniques	Thematic analysis	**The impact of HIV on WLHIV** (i) Psychological impact Feeling sad and bad due to contracting HIV Feeling embarrassed due to changes in physical appearance Psychological pressure and suicidal ideation due to stigma and discrimination by other family members (ii) Perceived stigma The belief about negative attitudes and behaviour of non-infected people towards PLHIV (iii) External stigma and discrimination From family members: - Social isolation by family members - Refusal by family members to share food and eating utensils - Avoidance by family members From healthcare professionals: - Being left untreated by healthcare professionals - Being isolated in untended and separated rooms in healthcare facilities
2. Chi, et al., 2010 [39]	Vietnam	(i) Qualitative design (ii) Methods: In-depth interview (iii) Aim: To explore their reflections, concerns and dilemmas related to reproductive decisions	(i) 13 WLHIV (ii) Marital status: 7 married 6 widowed (iii) Participants’ age: 30 years old and older (iv) Participant recruitment technique: Participants were recruited using convenient sampling technique Participants were part of a larger research project that investigated reproductive decisions among WLHIV	Thematic analysis	**The impact of HIV on WLHIV** (i) Psychological impact Fear of dying leading to mother choosing to terminate pregnancy Feeling worried about transmitting HIV to unborn baby Feeling worried about poor economic condition and inability to take care of the child (ii) Perceived stigma Fear of children being stigmatised and discriminated against within the community due to mother’s HIV status Fear of children being denied from school admission **The impact of HIV on families of WLHIV** (i) External stigma and discrimination against children at school Teacher told other children not to play with them. Children were placed at a separate desk by teacher and isolated from friends
3. de Souza, 2010 [33]	India	(i) Qualitative design (ii) Methods: In-depth Interview (iii) Aim: To understand women’s experience of power and powerlessness	(i) Two WLHIV (ii) Marital status: Remarried and widowed (iii) Participants’ age: 31 and 33 years of age. (iv) Participant recruitment technique: Participants were recruited using purposive sampling technique Participants were recruited from or part of a larger study identifying NGO practices to mobilize communities for HIV/AIDS prevention. These two women were chosen because they share similar biography and illness narratives.	Content analysis	**The impact of HIV on WLHIV** (i) External stigma and discrimination From parents and siblings From parents-in-law From community members
4. Fauk, et al., 2021 [12]	Indonesia	(i) Qualitative design (ii) Methods: In-depth Interview (iii) Aim: To explore perception of PLHIV about drivers of stigma and discrimination towards them within families, communities and healthcare settings	(i) 52 WLHIV and 40 MLHIV (ii) Marital status: Married Unmarried (divorced, widowed and never married) (iii) Participants’ age: Ranged between 18 and 60 years old (iv) Participant recruitment technique: Participants were recruited using the snowball sampling technique Only the results about the impact of HIV on the female participants are included.	Thematic analysis	**The impact of HIV on WLHIV** (i) Externa stigma and discrimination From family members and in-laws - Separation of and giving a sign to eating utensils - Separation of their children from them - Being labelled as sex workers From community members - Refusal of sharing foods and drinks they have touched - Being labelled as sex workers, people with bad behaviours and immoral From healthcare professionals - Delay of treatment - Having their HIV status being spread by healthcare professionals - Being labelled as sex workers
5. Halli, et al., 2017 [4]	India	(i) Quantitative design: Cross-sectional study (ii) Aim To examine HIV/AIDS-related stigma and discrimination in a high-HIV-prevalence district in India	(i) 633 married WLHIV (ii) Marital status: Married (iii) Participants’ age: Ranged between 15 and 29 years old. (iv) Participant recruitment technique: Participants were randomly selected from a unique cross-sectional quantitative study conducted among HIV-positive women in Bagalkot District of Karnataka, India.	Bivariate analysis and multivariate logistic regression models	**The impact of HIV on WLHIV** (i) Externa stigma and discrimination From parents and siblings From husband and husband’s family members From community members (friends and neighbours) Sociodemographic factors associated with stigma and discrimination from husband and husband’s family members against WLHIV: - Occupation and age of the husband - Higher age gap between spouses and poor household status - Older age of the husband and lower household economic status
6. Halimatusa’diah, 2019 [37]	Indonesia	(i) Qualitative design (ii) Methods: Semi-structured and unstructured interview (iii) Aim: To explore the struggle of WLHIV in dealing with the moralisation of HIV in Indonesia	(i) 33 WLHIV (ii) Marital status: Married and widowed (iii) Participants’ age: 18–45 years old (iv) Participant recruitment technique: Participants were recruited using purposive and snowball sampling techniques	Thematic analysis	**The impact of HIV on WLHIV** (i) Psychological impacts Feeling shocked, upset and angry - Due to their perceptions that they are good wives who only have sex with their husbands Self-blaming - Due to unprotected sex, they engaged in with multiple partners (ii) Externa stigma and discrimination From in-laws - Being chased out of late husband’s family house
7. Ismail, et al., 2018 [38]	Indonesia	(i) Qualitative design (ii) Methods: In-depth interview (i) Aim: To describe the concerns of women infected with HIV by their IDU husbands	(i) 12 WLHIV (ii) Marital status: Married (iii) Participants’ age: 20–35 years old (iv) Participant recruitment technique: Participants were recruited through a non-governmental organisation	Content analysis	**The impact of HIV on WLHIV** (i) Psychological impacts Feeling shocked, upset and angry - Due to being infected by their spouses Denying HIV infection - Due to their perceptions that they are good wives and do not have sex with other men Concern about their children’s future (ii) External stigma and discrimination From parents and in-laws - Separation of and giving a sign to eating utensils - Being accused of the death of their spouses: accusation of transmitting HIV to their spouses
8. Liamputtong, et al., 2009 [41]	Indonesia	(i) Qualitative design (ii) Methods: In-depth interview (iii) Aim: To examine community attitudes toward women living with HIV and AIDS at the present time from the perspectives of women in Thailand.	(i) 26 WLHIV (ii) Marital status: Married and unmarried (divorced, widowed, separated and never married) (iii) Participants’ age: 20–50 years old (iv) Participant recruitment technique: Participants were recruited using purposive sampling technique	Thematic analysis	**The impact of HIV on WLHIV** (i) Anticipated stigma The belief that people will discriminate against them if their HIV status is known to others (ii) Externa stigma and discrimination From community members (friends and neighbours) - Being labelled as bad women who have sex with many men From healthcare professionals - Being accused by nurses as the ones who transmitted HIV to their spouses From employers - Being asked by employers to leave their jobs due to HIV-related physical appearance (face turned black)
9. Mathew, et al., 2019 [42]	Thailand	(i) Qualitative design (ii) Methods: Semi-structured qualitative interview (iii) Aim To explore the perspectives of young adults living with HIV in Bangkok regarding the influence of stigma and discrimination in education, employment, health care, personal relationships, and perceptions of self	(i) 14 WLHIVs and 9 MLHIV. (ii) Marital status: Single (iii) Participants’ age: 15–24 years old (iv) Participant recruitment technique: Participants were recruited using convenient and purposive sampling techniques Only the results about the impact of HIV on the female participants are included.	Thematic analysis	**The impact of HIV on WLHIV** (i) Perceived stigma The belief that other people will dislike and show negative reactions if one’s HIV status is known The belief that one will be excluded or denied from a certain job because of one’s HIV status
10. Nguyen, et al., 2009 [40]	Vietnam	(i) Qualitative design (ii) Methods: In-depth interview (iii) Aim: To explore the experience of 30 WLHIV in Vietnam in accessing HIV-related postnatal care, the role of felt and enacted stigma in accessing services, and the effects of participation in a self-help group on utilization of available services.	(i) 30 married WLHIV (ii) Marital status: Married (iii) Participants’ age: Ranged between 25 and 35 years. (iv) Participant recruitment technique: Participants were recruited using convenient sampling technique Participants were recruited through healthcare facilities and referral of healthcare professionals	Thematic analysis	**The impact of HIV on WLHIV** (i) External stigma and discrimination From husband’s family (mother and sister) - Left by in-laws due to HIV-positive status of WLHIV - Asked by in-laws to go back to their parents - Avoided by in-laws From community members - Unaccepted and avoided by community members - Community members being cynical towards them From healthcare professionals - Left unattended and untreated by healthcare professionals in hospitals **The impact of HIV on families of WLHIV** (i) External stigma and discrimination From husband’s family (mother and sister) - Bowls and chopsticks were boiled by sister-in-laws - Refusal by in-laws to live together with them (ii) Family separation Their children are kept away from them by in-laws
11. Paxton, et al., 2005 [44]	India, Indonesia, Thailand, the Philippines	(i) Quantitative design: Cross-sectional study (ii) Aim: To develop an understanding of the nature, pattern and extent of AIDS-related discrimination in several Asian countries	(i) 764 PLHIV in four countries (India 302; Indonesia 42; Thailand 338; the Philippines 82) (ii) 348 respondents were female: 40% married 50% widowed 10% single (iii) 394 were male, six were transgender and for five sex was not recorded (iv) Participants’ age: Not reported (v) Participant recruitment technique: Participants were recruited using snowball sampling technique. Only results about WLHIV were used.	Chi square test	**The impact of HIV on WLHIV** (i) External stigma and discrimination From healthcare professionals - Refusal of treatment or a delay in the provision of healthcare - Being coerced into an HIV test - Undergoing mandatory testing while they were pregnant or because of the illness of a child - Being advised not to have children after diagnosis without giving information about prevention of mother-to-child transmission. - Being coerced into an abortion or sterilization after diagnosis - Losing or being denied private insurance once their HIV status was known and discriminated in relation to private insurance From community members - Being refused entry to, asked to leave or removed from a public establishment (including places of religious worship) - Changing their place of residence due to their HIV status - Being excluded from social functions due to their status - Being physically assaulted because of their status - Being required to disclose their HIV status in order to enter another country - Being excluded from usual activities - Being excluded from associations or clubs due to their status and/or restricted in their ability to meet with other PLHIV From family members - Being excluded from usual household activities such as cooking, sharing food or eating implements and sleeping in the same room as others. - Losing financial support from their spouse or other family members due to HIV status - Being chased her out of her home by mother in-law due HIV status At workplace - Loss of jobs - Change in job description or duties - Loss prospects for promotion **The impact of HIV on families of WLHIV** (i) Impact on children’s education children were denied admission into schools (ii) Family separation Children involuntarily taken away from them due to their HIV status Desertion/abandonment by their spouse because of their diagnosis
12. Qin, 2018 [8]	China	(i) Quantitative design: Cross-sectional study (ii) Aim To explore the psychological distress of HIV-infected pregnant women, and analyse the possible influencing factors	(i) 194 pregnant WLHIV (ii) Marital status: Married (iii) Participants’ age: The average age of the participants was 25.1 ± 5.8 years (iv) Participant recruitment technique: Participants were recruited using convenient sampling technique	Multiple linear regression analysis	**The impact of HIV on WLHIV** (i) Psychological impact Anxiety and depression Factors associated with psychological distress - Family misfortune, Medicaid, chronic disease or high-risk pregnancy, viral load, CD4 þT cell count, infection and confidentiality
13. Srivastava, et al., 2017 [34]	India	(i) Qualitative design (ii) Methods: Focus group discussion (iii) Aim: To explore perspectives of WLHIV on their mental health and the role and impact of Accredited Social Health Activity (ASHA) support on mental health after participation in a community-based intervention	(i) 16 WLHIV who were mothers (ii) Marital status: Not reported (iii) Participants’ age: Mean age was 32.6 (iv) Participant recruitment technique: They selected from a group of 34 WLHIV who participated in ASHA intervention	Content analysis	**The impact of HIV on WLHIV** (i) Psychological impact Depressed mood, anxiety and suicidal ideation Factors associated with psychological distress: - Experience of stigma and discrimination by neighbours and relatives - Experience of stigma and discrimination by healthcare professionals - Concerns about their children’s life and future
14. Subramaniyan, et al., 2013 [35]	India	(i) Qualitative design (ii) Methods: In-depth interview (iii) Aim: To explore the difficulties faced by rural HIV positive mothers during the intra-natal period.	(i) 21 WLHIV who were mothers (ii) Marital status: Not reported (iii) Participants’ age: Ranged between 20 and 39 years. (iv) Participant recruitment technique: Not reported	Content analysis	**The impact of HIV on WLHIV** (i) External stigma and discrimination From healthcare professionals - Avoidance of physical examination, rude behaviour such as the throwing of records at the face, discriminatory comments, unnecessary referrals and even refusal to provide intra-partum services - Leading to women concealing their HIV status, not accessing healthcare services or looking for healthcare services in other places where their HIV status is unknown to healthcare providers
15. Thomas, et al., 2009 [36]	India	(i) Qualitative design (ii) Methods: Focused group discussion (iii) Aim: To explore the perceptions and needs of mothers living with HIV to gain greater insights into the challenges they face in relation to their health seeking behavior, fears around disclosure, and issues related to stigma and discrimination.	(i) 60 WLHIV (ii) Marital status: Married and unmarried (iii) Participants’ age: Ranged between 23 and 42 years (iv) Participant recruitment technique: Participants were recruited from two maternity hospitals, a large sexually transmitted disease (STD) clinic and antiretroviral therapy (ART) clinic	Content analysis	**The impact of HIV on WLHIV** (i) External stigma and discrimination From spouses and in-laws - Separation of eating utensils - Refusal of sharing personal belongings From healthcare professionals - Avoidance of physical examination - Putting WLHIV in rooms with no facilities
16. Zhang, et al., 2022 [43]	China	(i) Qualitative and quantitative (mixed-method) design (ii) Methods: Structured questionnaire In-depth interview (iii) Aim To investigate stigmatization and social support of pregnant women with HIV or syphilis in eastern China	(i) 93 WLHIV (ii) 355 women living with syphilis (iii) Marital status: Not reported (iv) Participants’ age: Ranged between ≤ 20 and ≥ 35 years (v) Participant recruitment technique: Participants were recruited at a local women and children hospital	Descriptive analysis; *t*-tests; Chi Square tests; Content analysis	**The impact of HIV on WLHIV** (i) Psychological impacts Feeling terrible about their HIV status (ii) External stigma and discrimination From community members From healthcare professionals - Stigmatising attitudes (talking about patients’ HIV status)
17. Yang, 2015 [18]	Cambodia	(i) Qualitative design (ii) Methods: In-depth interview (iii) Aim To explore women’s perspectives on their life changes after being infected with HIV by their husbands.	(i) 15 WLHIV (ii) Marital status: Married (iii) Participants’ age: Ranged between 30 and 42 years (iv) Participant recruitment technique: Participants were recruited using snowball sampling technique	(i) thematic analysis, (ii) analysis of episodes, and (iii) identification of paradigm cases	**The impact of HIV on families of WLHIV** (i) Family separation Children live with other families for better housing and due to mother’s sickness

## Data Availability

All data generated or analysed during this study or review are included in this published article.

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
