# Peer review of "Psychological and Social Impact of HIV on Women Living with HIV and Their Families in Low- and Middle-Income Asian Countries: A Systematic Search and Critical Review"

_ijerph, 2022, doi:10.3390/ijerph19116668_

Round 1

Reviewer 1 Report

I thank the authors for making the revisions to their manuscript.  It has strengthened the manuscript.  I recommend that it be accepted.

Author Response

We would like to thank the reviewer for providing valuable comments for the improvement of our manuscript at first round of the review.

Reviewer 2 Report

a). Avoid double use of HIV in the title.

b). Define abbreviations in the Abstract (LMIC and WLHIV), as well as in the text (NKF and LM).

c). Define in a concrete way the conceptualization of low and medium income families.

d). In the title and in the text, I'm sorry, it could be confused who has low and middle income, families or countries.

e). Figure 2 should be considered as a simplified conceptual model and not only as a conceptual model, since the study could even be considered to have other representations of broad conceptual models, even if they are not called that.

f). The study is presented as a systematic review, but could not be better considered as a qualitative meta-analysis.

g). Table 2 should be an annex, it is too broad and should be presented in the body of the document or separated into parts and each part should do its own analysis.

Author Response

This manuscript is a resubmission of an earlier submission. The following is a list of the peer review reports and author responses from that submission.

Round 1

Reviewer 1 Report

This is a good manuscript - well designed and make an important contribution.

The abstract is well written.  The introduction provides good statistical data.  But authors should provide definitions of middle and low income countries.  The literature review section needs more inclusion of current literature. The authors cite literature, but which countries were these studies carried out in?  When?  Some themes need to be discussed further - like stigma, economic insecurity, health, etc.

The rationale for a systematic review is not clear.  What is the purpose if the literature already exists.  What does this review achieve and why is it beneficial to the field - be explicit.

Methodology is well written. Could there be a table to summarize the JBI method?

The findings and discussion of findings are excellent.

Implications - need to specifically state what this review revealed that is different from the studies included.  What is new or relevant that is driving the implications?

Reviewer 2 Report

I read with great interest the research entitled "The impact of HIV on women living with HIV and their families in low- and middle-income countries: A systematic review ".

The importance of the study is undeniable, as explained in the introduction, and many of its conclusions are interesting. However, it seems to me that this is a paper that in its current format cannot be published and that the corrections that, from my point of view, should be made do not allow a recommendation for changes, but a rejection.

The introduction is correct, but the scope of the term "impact of HIV" and the different bifurcations of "women living with HIV in developing countries" are not fully developed. Which brings us to two of the four main elements of criticism I find with this paper:

  1. The concept of "impact" is huge. And the articles that are selected open up a huge number of inclusion terms. We can understand that these will include many of the relevant articles on the subject, but not those that measure impact in different terms. The reality is that more than "impact", I would recommend talking about different areas (social, psychological, gender violence...) in which this is of interest.
  2. The heterogeneity of the countries included is enormous. They are united by their economic income, but have many other variables (culture, religion, politics...) that make the grouping of the results very complex to convey.

In addition, I believe that for a systematic review, results from very different methods are pooled. Not only are qualitative and quantitative results combined, but among the quantitative results, those from cross-sectional studies are grouped together (shown in different columns, but the conclusions group them together) with those from prospective studies.

And finally, these results are shown in the form of "areas" of impact, but without measures of association. We know the relevant aspects, but we are left without knowing (neither in the tables nor in the results) the strength of these associations.

In short, and with the utmost constructive spirit, I think it is an article of interest. It draws many interesting conclusions and creates a conceptual map that I consider adequate. But the mix of countries, of what impact can mean and of methodologies makes me think that it would have been better to present this article divided into parts. Or thought more for a different review article

The aspects that are not commented on are either adequate or minor in relation to the reasons given.

Hoping that these comments may be of help, best regards.